# Clinical Significance of Toxigenic *Clostridioides difficile* Growth in Stool Cultures during the Era of Nonculture Methods for the Diagnosis of *C. difficile* Infection

Ching-Chi Lee,[a,b] Jen-Chieh Lee,[a] Chun-Wei Chiu,[c] Pei-Jane Tsai,[d,e,f] Wen-Chien Ko,[a,g] Yuan-Pin Hung[a,c]

[a]Department of Internal Medicine, National Cheng Kung University Hospital, College of Medicine, National Cheng Kung University, Tainan, Taiwan
[b]Clinical Medicine Research Center, National Cheng Kung University Hospital, College of Medicine, National Cheng Kung University, Tainan, Taiwan
[c]Department of Internal Medicine, Tainan Hospital, Ministry of Health and Welfare, Tainan, Taiwan
[d]Department of Medical Laboratory Science and Biotechnology, National Cheng Kung University, College of Medicine, Tainan, Taiwan
[e]Department of Pathology, National Cheng Kung University Hospital, Tainan, Taiwan
[f]Centers of Infectious Disease and Signaling Research, National Cheng Kung University, Tainan, Taiwan
[g]Department of Medicine, College of Medicine, National Cheng Kung University, Tainan, Taiwan

**ABSTRACT** The importance of the detection of relevant toxins or toxin genes to diagnose *Clostridioides difficile* infection (CDI) or the prediction of clinical outcomes of CDI has been emphasized in recent years. Although stool culture of *C. difficile* is not routinely recommended in the era of nonculture methods as the preferred tools for CDI diagnosis, the clinical significance of toxigenic *C. difficile* growth (tCdG) in stool cultures was analyzed. A clinical study was conducted in medical wards of Tainan Hospital, Ministry of Health and Welfare, in southern Taiwan. Diarrheal adults with fecal glutamate dehydrogenase and *C. difficile* toxin between January 2013 and April 2020 were included. Of the 209 patients with CDI, 158 (75.6%) had tCdG found in stool cultures, and the rest (51, 24.4%) had no tCdG in stool. Only prior ceftazidime or ceftriaxone therapy was independently associated with no tCdG in stool (odds ratio [OR] 2.17, $P = 0.02$). Compared to the patients with tCDG in stool, those without tCdG in stool experienced treatment success more often (97.1% versus 67.0%, $P < 0.001$) if treated with metronidazole or vancomycin but had a similar in-hospital mortality or recurrence rate. In the multivariate analysis among 114 patients with CDI treated with metronidazole or vancomycin, treatment success was independently associated with no tCdG in stool (OR 12.7, $P = 0.02$). Despite the limited utility of stool cultures in CDI diagnoses, no tCdG in stool culture heralds a favorable therapeutic outcome among adults with CDI treated with metronidazole or vancomycin.

**IMPORTANCE** The importance of detecting toxins or toxin genes when diagnosing *Clostridioides difficile* infections (CDIs) or predicting the severity and outcomes of CDI has been emphasized in recent years. Although the yielding of *C. difficile* from stool cultures might implicate higher bacterial loads in fecal samples, in an era of nonculture methods for the standard diagnosis of CDIs, clinical significance of positive stool cultures of toxigenic *C. difficile* was analyzed in this study. Despite the limited ability of stool cultures in CDI diagnoses, no yielding of *C. difficile* growth might predict the successful CDI therapy.

**KEYWORDS** *Clostridioides difficile* infection, metronidazole, *Clostridioides difficile*, enzyme immunoassay, stool culture, successful therapy

Address correspondence to Yuan-Pin Hung, yuebin16@yahoo.com.tw, or Wen-Chien Ko, winston3415@gmail.com.

*C*lostridioides difficile infection (CDI) is a common community-acquired or health care-associated intestinal infection, with effects ranging from mild diarrhea to pseudomembranous colitis, toxic megacolon, or even death, with a mortality rate of up to 25% to 40% (1–9). Metronidazole and vancomycin therapy can both achieve substantial clinical cure rates

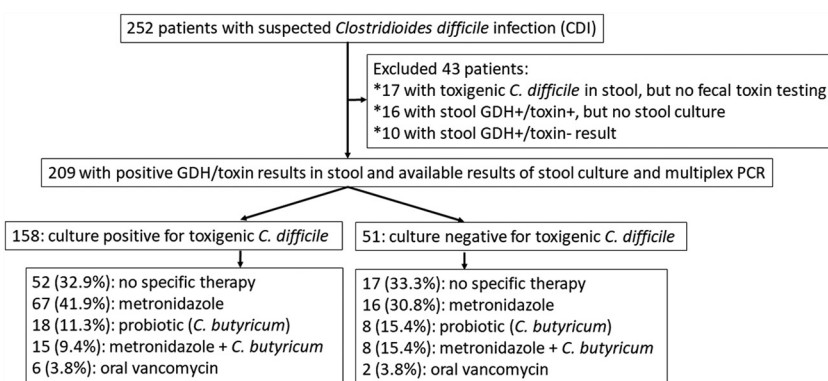

FIG 1 Patient flowchart of *Clostridioides difficile* infection, stratified by *C. difficile* growth in fecal samples.

among patients with mild to moderate CDI; however, for those with severe CDI, vancomycin therapy can provide a better clinical cure rate than metronidazole therapy (10). In the latest clinical guidelines updated by the Infectious Diseases Society of America (IDSA) and the Society for Healthcare Epidemiology of America (SHEA), oral vancomycin is preferred for either mild to moderate or severe CDI (11). Either metronidazole or vancomycin therapy has been associated with considerable treatment failure and recurrence rates (12). Although some probiotics, such as *Clostridium butyricum*, revealed *C. difficile*-inhibiting ability, there was insufficient evidence to suggest the routine use of probiotics for the prevention or treatment of CDI (13, 14).

The recognized virulence factors of *C. difficile* are toxins A and B (12), which are transcribed from a pathogenicity locus consisting of five genes: two toxin genes, namely, *tcdA* (toxin A) and *tcdB* (toxin B), and three regulatory genes. One of the latter genes, *tcdC*, is a negative regulator of toxin production (12, 15). Since the pathogenic role of *C. difficile* toxins has been established, the clinical diagnosis of CDI in principle relies on nucleic acid amplification tests (NAATs) to detect toxin genes (*tcdA* and/or *tcdB*) or enzyme immunoassays (EIAs) to detect glutamate dehydrogenase (GDH) and toxin A/B (16). Moreover, the impact of the presence of *C. difficile* toxins or toxin genes on diagnosing CDI or predicting the disease severity or outcomes of CDI has been emphasized recently (11). Although stool cultures of *C. difficile* are more sensitive than EIAs for detecting GDH of *C. difficile* or toxins in fecal samples, the drawbacks of stool cultures include the need for anaerobic incubators, time-consuming workflows, and the detection of both toxigenic and nontoxigenic *C. difficile* isolates. Thus, the above microbiological issues of stool culture vastly decrease its diagnostic efficacy for CDI (17–19). Therefore, stool culture is currently suggested for epidemiological studies or to detect pathogens of antibiotic-associated diarrhea other than *C. difficile* (20, 21).

With the quantitative measurement of toxigenic *C. difficile* bacterial loads in fecal samples by NAATs, a correlation between the growth of *C. difficile* from perianal or stool samples and the cycle threshold ($C_T$) to PCR positivity has been established, which generally is correlated with the *C. difficile* bacterial load (22). Therefore, we considered that the growth of *C. difficile* in stool cultures, in part, might represent higher bacterial loads in fecal samples. Accordingly, in an era of nonculture methods for the rapid diagnosis of CDI, we aim to investigate the clinical feasibility of toxigenic *C. difficile* stool cultures for predicting the therapeutic outcome of antimicrobial therapy.

## RESULTS

In the study period, 252 patients were clinically diagnosed with CDI, but only 209 patients with unexplained diarrhea met the inclusion criterion of the presence of both GDH and *C. difficile* toxin A/B detected by EIA in stool and were included for further analyses (Fig. 1). Of them, tCdG in stool was noted in 158 (75.6%) patients, and no

**TABLE 1** Underlying diseases and prior antibiotic or medication exposure in patients with *Clostridioides difficile* infection (CDI), stratified by the results of stool culture

| Variables (case/isolate no.) | Value for stool culture for tCd[a] | | | |
| | Total, $n = 209$ | Positive, $n = 158$ | Negative, $n = 51$ | P value |
| --- | --- | --- | --- | --- |
| Age, yrs | 75.8 ± 12.5 | 76.2 ± 12.1 | 74.6 ± 14.0 | 0.47 |
| Sex, male | 104 (49.8) | 78 (49.4) | 26 (51.0) | 0.87 |
| | | | | |
| Underlying disease | | | | |
| Hypertension | 132 (63.2) | 94 (59.5) | 38 (74.5) | 0.07 |
| Diabetes mellitus | 96 (45.9) | 73 (46.2) | 23 (45.1) | 1.00 |
| Chronic kidney disease | 87 (41.6) | 65 (41.1) | 22 (43.1) | 0.87 |
| Old stroke | 79 (37.8) | 59 (37.3) | 20 (39.2) | 0.87 |
| Dementia | 56 (26.8) | 45 (28.5) | 11 (21.6) | 0.37 |
| Coronary artery disease history | 37 (17.7) | 27 (17.1) | 10 (19.6) | 0.68 |
| Congestive heart failure | 29 (13.9) | 24 (15.2) | 5 (9.8) | 0.49 |
| Parkinsonism | 27 (12.9) | 21 (13.3) | 6 (11.8) | 1.00 |
| Malignancy | 26 (12.4) | 20 (12.7) | 6 (11.8) | 1.00 |
| Chronic obstructive pulmonary disease | 19 (9.1) | 11 (7.0) | 8 (15.7) | 0.09 |
| Liver cirrhosis | 6 (2.9) | 4 (2.5) | 2 (3.9) | 0.64 |
| | | | | |
| Recent medication within 1 mo before CDI onset | | | | |
| Antimicrobial therapy | | | | |
| Cephalosporins | 108 (51.7) | 79 (50.0) | 29 (56.9) | 0.42 |
| Cefazolin, i.v. | 3 (1.4) | 2 (1.3) | 1 (2.0) | 0.57 |
| Cefuroxime, i.v./o | 17 (8.1) | 14 (8.9) | 3 (5.9) | 0.77 |
| Ceftazidime or ceftriaxone, i.v. | 66 (31.6) | 43 (27.2) | 23 (45.1) | 0.02 |
| Cefepime, i.v. | 30 (14.4) | 27 (17.1) | 3 (5.9) | 0.06 |
| Penicillins | 23 (11.0) | 17 (10.8) | 6 (11.8) | 0.80 |
| Carbapenem, i.v. | 38 (18.2) | 29 (18.4) | 9 (17.6) | 1.00 |
| Fluoroquinolones, i.v./o | 10 (4.8) | 9 (5.7) | 1 (2.0) | 0.46 |
| Glycopeptide, i.v. | 25 (12.0) | 20 (12.7) | 5 (9.8) | 0.80 |
| Tigecycline, i.v. | 3 (1.4) | 2 (1.3) | 1 (2.0) | 0.57 |
| Doxycycline, o | 2 (1.0) | 2 (1.3) | 0 | 1.00 |
| Proton pump inhibitors, i.v./o | 52 (24.9) | 38 (24.1) | 14 (27.5) | 0.71 |
| H2-receptor antagonists, i.v./o | 32 (15.3) | 27 (17.1) | 5 (9.8) | 0.27 |
| Steroid, i.v./o | 53 (25.4) | 41 (25.9) | 12 (23.5) | 0.85 |

[a]Data are presented as patient numbers (%) or means ± standard deviations. tCd, toxigenic *C. difficile*; i.v./o, intravenous/oral.

tCdG was noted in 51 (24.4%) patients. None of the included patients had the growth of nontoxigenic *C. difficile* isolates in stool.

Of the 158 patients with tCdG in stool, 52 (32.9%) received no specific therapy, 67 (41.9%) received oral or intravenous metronidazole, 18 (11.3%) received *C. butyricum* probiotics, 15 (9.4%) received metronidazole plus *C. butyricum*, and 6 (3.8%) received oral vancomycin, as shown in Figure 1. Of the 51 patients without tCdG in stool, 17 (33.3%) received no specific therapy, 16 (30.8%) received oral or intravenous metronidazole, 8 (15.4%) received *C. butyricum*, 8 (15.4%) received oral or intravenous metronidazole plus *C. butyricum*, and 2 (3.8%) received oral vancomycin. Among 69 patients without specific anti-*C. difficile* therapy, their diarrhea all resolved after discontinuing the offending antibiotics.

Compared to those without tCdG in stool, the patients with tCdG had more often been previously treated with ceftazidime or ceftriaxone (45.1% versus 27.2%, $P = 0.02$). In addition, those with tCdG more often had underlying hypertension (74.5% versus 59.5%, $P = 0.07$) or chronic obstructive pulmonary disease (15.7% versus 7.0%, $P = 0.09$), although the differences were not statistically significant (Table 1). In terms of age, sex, other underlying diseases, prior antibiotic therapy (including oral doxycycline and intravenous tigecycline), or prior exposure to steroids or suppressors of gastric acid secretion, there were no differences between the two groups.

Between patients without tCdG and those with tCdG in stool, there were no significant differences in the proportions of severe CDIs (43.1% versus 32.9%, $P = 0.24$), leukocytosis

**TABLE 2** Laboratory characteristics and antimicrobial therapy in patients with *Clostridioides difficile* infection (CDI), stratified by the results of stool culture

| Variables | Value for stool culture for tCd[a] | | | |
|---|---|---|---|---|
| | Total, *n* = 209 | Growth, *n* = 158 | No growth, *n* = 51 | *P* value |
| Leukocyte count, cells/ml | 12.5 ± 7.1 | 12.5 ± 7.4 | 12.5 ± 6.5 | 0.98 |
| >15,000 cells/ml | 60 (28.7) | 43 (27.2) | 17 (33.3) | 0.48 |
| | | | | |
| Serum creatinine of >1.5 mg/liter | 23 (11.0) | 17 (10.8) | 6 (11.8) | 0.80 |
| Severe CDI by IDSA/SHEA criteria[b] | 74 (35.4) | 52 (32.9) | 22 (43.1) | 0.24 |
| | | | | |
| Drug therapy for CDI | | | | |
| Metronidazole, i.v./o | 107 (51.2) | 83 (52.5) | 24 (47.1) | 0.52 |
| Probiotic (*Clostridium butyricum*), o | 51 (24.4) | 35 (22.2) | 16 (31.4) | 0.19 |
| Vancomycin, o | 8 (3.8) | 6 (3.8) | 2 (3.9) | 1.00 |

[a]Data are presented as patient numbers (%) or means ± standard deviations. tCd, toxigenic *C. difficile*; i.v./o, intravenous/oral.
[b]Leukocyte count of ≥15,000 cells/ml or serum creatinine of >1.5 mg/dl.

(white blood cells [WBC] > 15,000 cells/ml: 33.3% versus 27.2%, *P* = 0.48), or serum creatinine level of >1.5 mg/liter (11.8% versus 10.8%, *P* = 0.80) (Table 2). In addition, the percentages of metronidazole therapy (47.1% versus 52.5%, *P* = 0.52), vancomycin (3.9% versus 3.8%, *P* = 0.52), and *C. butyricum* probiotic (31.4% versus 22.2%, *P* = 0.19) were similar between the two patient groups.

Clinical parameters with a *P* value of <0.15 in the univariate analysis, including prior ceftazidime or ceftriaxone therapy, underlying hypertension, and chronic obstructive pulmonary disease, were selected for the multivariate analysis. Prior cefepime therapy, although with a *P* value of <0.15, was confounded by prior ceftazidime or ceftriaxone therapy and therefore was not included in the multivariate analysis. A multivariate analysis for clinical predictors associated with no tCdG in stool in 209 patients with CDIs showed one independent factor, prior ceftazidime or ceftriaxone therapy (odds ratio [OR] 2.17; 95% confidence interval [CI] 1.12 to 4.22; *P* = 0.02) (Table 3).

Compared to CDI patients with tCdG in stool, those without tCdG were more likely to experience treatment success (97.1% versus 67.0%, *P* < 0.001) but had similar in-hospital mortality rates (25.5% versus 21.5%, *P* = 0.53) and CDI recurrence rates (9.8% versus 5.7%, *P* = 0.34) (Table 4). Of the 209 patients with CDI, 114 (54.5%) received standard antimicrobial therapy, including metronidazole or vancomycin, and those experiencing treatment success were more likely to have no tCdG in stool (30.5% versus 3.1%, *P* = 0.001), to have hypertension (74.4% versus 50.0%, *P* = 0.02), and to be older (mean: 78.3 years versus 72.6 years, *P* = 0.03) or female (53.7% versus 37.5%, *P* = 0.15), as shown in Table 5. In the multivariable analysis, no tCdG in stool was independently associated with treatment success among 114 patients receiving metronidazole or vancomycin therapy (OR 12.70; 95% CI 1.57 to 102.89; *P* = 0.02) (Table 6).

## DISCUSSION

No toxigenic *C. difficile* growth in stool cultures was associated with treatment success for hospitalized adults with CDI in our study if metronidazole or vancomycin was prescribed. This result might be reasonably related to a lower *C. difficile* bacterial burden in stool. Generally, lower fecal levels of *C. difficile* toxins have been linked to milder disease and lower mortality

**TABLE 3** Multivariate analyses of clinical variables associated with no growth of toxigenic *Clostridioides difficile* in stool cultures among 209 patients with *C. difficile* infection

| Clinical variables | Odds ratio | 95% confidence interval | *P* value |
|---|---|---|---|
| Hypertension | 2.02 | 0.98–4.15 | 0.06 |
| Prior ceftazidime or ceftriaxone therapy | 2.17 | 1.12–4.22 | 0.02 |
| Chronic obstructive pulmonary disease | 2.31 | 0.86–6.22 | 0.10 |

**TABLE 4** Clinical outcomes of 209 patients with *Clostridioides difficile* infection, stratified by the results of stool culture

| | Value for stool culture for tCd[a] | | |
|---|---|---|---|
| Clinical variables | Growth, *n* = 158 | No growth, *n* = 51 | *P* value |
| Treatment success[b] (140[c]) | 71 (67.0) | 33 (97.1) | <0.001 |
| Hospitalization duration, days | 29.8 ± 19.4 | 26.2 ± 17.2 | 0.33 |
| In-hospital mortality | 34 (21.5) | 13 (25.5) | 0.53 |
| Recurrence | 9 (5.7) | 5 (9.8) | 0.34 |

[a]Clinical variables are expressed as patient numbers (%) or means ± standard deviations; tCd, toxigenic *C. difficile*.
[b]Resolution of diarrhea within six days of the indicated therapy without a change in the therapeutic regimen.
[c]Available case number.

rates (23–26), and a high toxin level (>2,500 ng/ml) is associated with a higher mortality rate in patients with CDI (23). Additionally, the median fecal toxin levels have been consistently correlated with diarrheal frequencies in patients with CDI (24). It is reasonable to consider that the fecal toxin level is one of the parameters to predict treatment efficacies. Likewise, the growth of *C. difficile* in stool cultures indicates higher bacterial loads, which are supported by the quantitation of *C. difficile* bacterial loads in fecal samples by NAAT (22). There was a significant correlation between the amount of the *tcdB* gene reflected by the PCR $C_T$ values and the toxin levels detected by EIA with fecal bacterial loads (27). However, unlike the clinical significance of the fecal toxin levels that had been emphasized, the correlation between fecal *C. difficile* bacterial loads and disease severity was discordant (28, 29). A high *C. difficile* load has been regarded as a microbiological predictor of a poor outcome (29), but in another study regarding patients with CDI, a higher fecal load of *C. difficile*, either in spores or vegetative cells, was not correlated with more severe diarrhea as assessed by the Bristol stool scale (28). Thus, the linkage of fecal *C. difficile* bacterial loads and CDI severity needs further evaluation.

In addition to the prediction potential of the therapeutic outcomes of CDI in our study, time-consuming conventional stool cultures have been reported to provide higher sensitivity than EIAs for detecting fecal *C. difficile* toxins (17, 18). In a clinical

**TABLE 5** Underlying diseases of 114 patients receiving metronidazole or vancomycin therapy for *Clostridioides difficile* infection (CDI), stratified by treatment outcome[a]

| | Values for treatment that was: | | |
|---|---|---|---|
| Variables | Unsuccessful, *n* = 32 | Successful, *n* = 82 | *P* value |
| Age, yrs | 72.6 ± 12.5 | 78.3 ± 11.9 | 0.03 |
| Sex, male | 20 (62.5) | 38 (46.3) | 0.15 |
| No growth of toxigenic *C. difficile* in stool | 1 (3.1) | 25 (30.5) | 0.001 |
| | | | |
| Underlying disease | | | |
| Hypertension | 16 (50.0) | 61 (74.4) | 0.02 |
| Diabetes mellitus | 11 (34.4) | 39 (47.6) | 0.22 |
| Chronic kidney disease | 15 (46.9) | 29 (35.4) | 0.29 |
| Old stroke | 12 (37.5) | 32 (39.0) | 1.00 |
| Dementia | 6 (18.8) | 21 (25.6) | 0.62 |
| Congestive heart failure | 5 (15.6) | 12 (14.6) | 1.00 |
| Malignancy | 3 (9.4) | 15 (18.3) | 0.39 |
| Parkinsonism | 3 (9.4) | 10 (12.2) | 1.00 |
| Coronary artery disease history | 4 (12.5) | 11 (13.4) | 1.00 |
| Chronic obstructive pulmonary disease | 2 (6.3) | 6 (7.3) | 1.00 |
| Liver cirrhosis | 1 (3.2) | 2 (2.4) | 1.00 |
| | | | |
| Severe CDI by IDSA/SHEA criteria | 37 (38.1) | 16 (37.2) | 1.00 |
| | | | |
| Antimicrobial therapy for CDI | | | |
| Metronidazole, i.v./o | 29 (90.6) | 78 (95.1) | 0.40 |
| Vancomycin, o | 3 (9.4) | 5 (6.1) | 0.68 |

[a]Data are presented as patient numbers (%) or means ± standard deviations. i.v./o, intravenous/oral.

**TABLE 6** Multivariate analysis of the variables associated with treatment success among 114 patients receiving metronidazole or vancomycin therapy for *Clostridioides difficile* infection

| Variables | Odds ratio | 95% confidence interval | P value |
|---|---|---|---|
| No growth of toxigenic *C. difficile* in stool | 12.70 | 1.57–102.89 | 0.02 |
| Hypertension | 2.03 | 0.81–5.07 | 0.13 |
| Age | 1.04 | 1.00–1.08 | 0.06 |
| Sex, male | 0.60 | 0.24–1.50 | 0.27 |

evaluation of a three-step diagnostic algorithm, stool bacteriological cultures provided higher sensitivity but lower specificity than an enzyme-linked fluorescent immunoassay alone (sensitivity and specificity: 92.8% and 93.3% versus 63.3% and 96.7%, respectively) (18). In another study, toxigenic stool cultures recovered 24.4% of stool samples harboring toxigenic *C. difficile* isolates that were missed by a fecal cytotoxin EIA (17). Compared with real-time PCR, the culture-based method using a commercial chromogenic medium could detect an additional 9% of positive specimens (30). Although real-time PCR has been suggested as the confirmatory tool when the results of GDH tests and toxin EIAs are inconsistent, according to the IDSA/SHEA guidelines, the higher cost of commercial real-time PCR tests means that the tests are not often available in resource-limited settings. Stool cultures are more cost effective than commercial real-time PCR tests and could be applied as a complementary test in the presence of discordant results of GDH testing and toxin EIAs. Accordingly, we believe that in the modern era of nonculture methods for the clinical diagnosis of CDI, the yields of toxigenic stool cultures are shown to have prognostic significance in the setting of conventional antimicrobial therapy for CDI.

Prior exposure to third-generation cephalosporins, such as ceftriaxone or ceftazidime, was associated with a lower yield rate of *C. difficile* from stool cultures in our study. The finding did not hint at third-generation cephalosporins as a prophylactic or therapeutic choice for CDIs. In contrast, third-generation cephalosporins have been positively correlated with the development of CDI (31). However, third-generation cephalosporins may promote intestinal colonization with cephalosporinase-producing *Bacteroides* strains, which preserve colonization resistance against vancomycin-resistant *Enterococcus* or *C. difficile* (32). The significance of disturbed gut microbiota following third-generation cephalosporins and the lower recovery rate of toxigenic *C. difficile* from stool cultures warrant more clinical investigations.

There were some limitations in this study. First, the association of no tCdG in stool and treatment success in our study was hypothesized to be related to a low *C. difficile* bacterial burden in stool. However, there is a lack of microbiological evidence supporting the correlation of tCdG and bacterial loads in stool. Second, other factors potentially affecting the yield rate of *C. difficile* stool cultures, such as the transport time from beds to bench, were not analyzed. Third, treatment success was arbitrarily defined as the resolution of diarrhea within 6 days without adjustment of therapeutic regimens and a survival outcome. Other factors affecting diarrheal duration, such as dietary habits or concurrent medications, were not considered in this study. Finally, fecal *C. difficile* toxin was qualitatively detected using a commercial EIA kit. The quantified measurement of *C. difficile* toxin was not available in our clinical laboratory, although fecal *C. difficile* toxin levels are often referred to as one of the prognostic factors of CDI (23–26).

In conclusion, no growth of toxigenic *C. difficile* in stool among the patients with positive results of both GDH tests and toxin A/B EIAs is independently associated with treatment success by metronidazole or vancomycin for CDI.

## MATERIALS AND METHODS

A clinical study was conducted in the medical wards of the Tainan Hospital, Ministry of Health and Welfare, in southern Taiwan between January 2013 and April 2020 and was approved by the institutional review board of National Cheng Kung University Hospital, Taiwan (approval number: B-ER-107-362). In general, CDIs were diagnosed as those suffering from diarrhea without alternative explanations and with *C. difficile* toxin detected in fecal samples, as defined previously (33). Diarrhea was defined as at least three unformed bowel movements per day for at least 2 days. In the study hospital, the presence of *C. difficile*

toxin A/B in stool samples relied on positive results of both glutamate dehydrogenase (GDH) and toxin A+B using an enzyme immunoassay (Abbott, Santa Clara, USA). Fecal samples were sent for *C. difficile* cultures at the discretion of attending physicians on cycloserine-cefoxitin-fructose agar (CCFA), which was incubated anaerobically for 24 to 48 h. After *C. difficile* isolates were isolated, genomic DNA was extracted with a genomic DNA minikit (Geneaid, Ltd., Taiwan) to identify toxigenic isolates. Multiplex PCR was used to detect *tcdA*, *tcdB*, *cdtA*, *cdtB*, and *tcdC* deletions in *C. difficile* isolates, as described previously (34). Adult patients with unexplained diarrhea and positive EIA results of GDH and toxin A/B in fecal samples and available stool culture and multiplex PCR results were included in the present study. Patients without GDH/toxin or *C. difficile* stool culture data were excluded. The included patients with CDI were further categorized into two groups, i.e., those with and those without toxigenic *C. difficile* growth (tCdG) in stool cultures.

Clinical data, including age, nasogastric tube use, and underlying diseases, were obtained from electronic medical records. An estimated glomerular filtration rate (eGFR) of <60 ml/min/1.73 m$^2$ for at least 3 months was considered to indicate chronic kidney disease (CKD) (35). Medications, including antibiotics, proton pump inhibitors, histamine H2-receptor antagonists, steroids, or a probiotic of *C. butyricum* (Miyarisan, Miyarisan Pharmaceutical, Japan), prescribed during hospitalization were recorded. The cephalosporin category included cefazolin, cefuroxime, cefmetazole, ceftriaxone, ceftazidime, or cefepime. The penicillin category included penicillin derivatives (penicillin, oxacillin, or piperacillin) and beta-lactam/beta-lactamase inhibitors (amoxicillin-clavulanic acid, ampicillin/sulbactam, or piperacillin-tazobactam). The carbapenem category included ertapenem, imipenem, or meropenem, and the glycopeptide category included vancomycin or teicoplanin. Tigecycline or doxycycline therapy within hospitalization before the onset of CDI was also recorded.

The severity score of CDI was graded according to the Clinical Practice Guidelines in 2017 by IDSA and SHEA. A patient with a leukocyte count of ≤15,000 cells/ml and a serum creatinine level of <1.5 mg/dl was regarded as having mild to moderate CDI and otherwise severe CDI (11). The duration of hospitalization preceding CDI was the period from admission to CDI onset, and only the first CDI episode was included. A modified definition of treatment success was as follows: the resolution of diarrhea within 6 days of antimicrobial therapy, without the need to change the therapeutic regimen, and survival at the end of antimicrobial therapy (10). Recurrence was defined as relapsing diarrhea with *C. difficile* toxin or *tcdB*-carrying *C. difficile* isolates detected in stool by day 21 after initial treatment success (10).

Statistical analysis was performed by statistical software (IBM SPSS, version 22.0). Continuous data are expressed as the means ± standard deviations. The $\chi^2$ test or Fisher's test was used for categorical variables, and Student's *t* test was used for continuous variables. A two-tailed *P* value of less than 0.05 was considered to be statistically significant. The Bonferroni correction was applied for multiple comparisons. Parameters with *P* values of less than 0.15 were selected for multivariate analysis. Multivariate analyses were performed using the Hosmer–Lemeshow test for goodness of fit for logistic regression models.

## ACKNOWLEDGMENTS

C.-W.C., P.-J.T., C.-C.L., W.-C.K., and Y.-P.H. designed the experiments, performed the experiments, analyzed the data, and participated in the writing of the manuscript. All authors read and agreed to the published version of the manuscript. The present study was supported by research grants from the Ministry of Science and Technology (MOST 108-2321-B-006-004, 109-2314-B-006-089-MY3, MOST 108-2320-B-006-043-MY3 and 110-2314-B-675-001) and the National Cheng Kung University Hospital (NCKUH-11004029), Taiwan.

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
