## [Reviewer comments · Microbiology Spectrum]

**Microbiology
Spectrum**

Clinical significance of toxigenic *Clostridioides difficile* growth in stool cultures during the era of nonculture methods for the diagnosis of *C. difficile* infection

Ching-Chi Lee, Jen-Chieh Lee, Chun-Wei Chiu, Pei-Jane Tsai, Wen-Chien Ko, and Yuan Pin Hung

Corresponding Author(s): Yuan Pin Hung, Tainan Hospital, Department of Health, Executive Yuan

Review Timeline:

Submission Date:	July 7, 2021
Editorial Decision:	July 19, 2021
Revision Received:	July 22, 2021
Editorial Decision:	August 7, 2021
Revision Received:	August 17, 2021
Accepted:	September 7, 2021

Editor: N. Esther Babady

Reviewer(s): The reviewers have opted to remain anonymous.

Transaction Report:

DOI: <https://doi.org/10.1128/Spectrum.00799-21>

July 19, 2021

Dr. Yuan Pin Hung
Tainan Hospital, Department of Health, Executive Yuan
Internal medicine
Tainan
Taiwan

Re: Spectrum00799-21 (Clinical significance of the growth of toxigenic *Clostridioides difficile* in stool cultures during the era of non-culture methods for CDI diagnoses)

Dear Dr. Yuan Pin Hung:

Thank you for submitting your manuscript to Microbiology Spectrum. When submitting the revised version of your paper, please provide (1) point-by-point responses to the issues raised by the reviewers as file type "Response to Reviewers," not in your cover letter, and (2) a PDF file that indicates the changes from the original submission (by highlighting or underlining the changes) as file type "Marked Up Manuscript - For Review Only". Please use this link to submit your revised manuscript - we strongly recommend that you submit your paper within the next 60 days or reach out to me. Detailed information on submitting your revised paper are below.

Link Not Available

Sincerely,

N. Esther Babady

Journals Department
Reviewer comments:

Reviewer #1 (Comments for the Author):

I read with interest the manuscript submitted by Lee et al. The paper fits within the scope of microbiology spectrum and is of sound technical quality and of use to the community. However, there are a few concerns which I present below.

Major comments:

The inclusion criteria remain unclear. Were patients with positive growth on CCFA only included if they were toxin positive by PCR? Or did they also have to be GDH / toxin EIA positive? Unclear how the final group (n=158) was selected based on the laboratory work up. I would recommend authors to clarify this in the methods as this central to transparency and reproducibility.

Authors should include more detailed models about MV model build such as how variables were selected for inclusion (looks like by univariate p value cut offs?) where any other variables of clinical significance included? How was the model then assessed (one common method is by Hosmer-Lemeshow test, a statistical test for goodness of fit for logistic regression models)

Line 165: Were patients who received no active treatment excluded? Would include actual numbers with the percentages that are presented in the results (eg _% (_/_)). Would recommend restricting this comparison to patients who received active C. difficile treatment as recommended by IDSA guidelines (metronidazole and vancomycin). Similar to the subsequent MV model analysis.

The writing style makes the manuscript hard to comprehend and follow at time. This is most pronounced in the discussion section. Would recommend the authors review discussion section for clarity. It is customary to have the main findings of the paper summarized and put into context in the opening paragraph of the discussion. Would suggest the authors do this to increase readability of the discussion.

Minor comments

In figure one the framing of the exclusion criteria is unclear. Specifically, the wording "rely on

Methods: Was stool culture done as routine clinical work? Or was this done specifically as part of this study? Would recommend the authors clarify.

In the abstract and throughout the manuscript would consider rewording the phrase "no yielding".

Line 88: Would recommend rewording this statement. It is unclear to me what the authors mean by "vast correlation between the positive yielding of C. difficile from perianal samples and stool sample with the cycle thresholds(Ct) to PCR positivity". In the referenced study there was a weak correlation between peri-anal swab CT and PCR positivity and stool CT values and PCR positivity.

Line 207: Agree with the authors that the toxin EIA has unsatisfactory sensitivity as a stand-alone test for C. difficile diagnosis. However, with the advent and widespread use of NAAT toxins, I'm not sure there exist a role for routine C. difficile culture in the clinical settings and would recommend the authors reconsider this statement. Especially because the authors do not present any data in their

own study to support this.

One limitation in the study is the omission of other antibiotic classes, including clindamycin which has a strong association with *C. difficile* occurrence and tetracyclines (eg: doxycycline or tigecycline) which may have protective function against *C. difficile* occurrence.

Given the link between toxin level and outcomes, could the findings here be driven by toxin level vs. mere *c. diff* burden as measured by ability to culture. This is not explored in the study although brought up in the discussion, and should likely be included as a limitation or expanded on in the discussion

Reviewer #2 (Comments for the Author):

This is a well thought out study with an intriguing hypothesis that does indeed merit further study. The authors' primary conclusion that the culture negative group responded more to therapy has some major flaws however.

The methodology description needs more detail, in order to lend credence to the conclusions which cannot be supported without additional information. The definition of positive *C. difficile* diagnoses is confusing. In the methods section, the authors outline that they used a multiplex PCR for the detection of toxin genes, but also a GDH and EIA assay for toxins. And yet this is not clarified in the flow chart in Figure 1 (what groups exactly did they include in their definition for toxin positive? What did they do with the discordants - PCR negative vs GDH/EIA positive or PCR positive vs GDH/EIA negative?).

Correlation with cycle thresholds of toxin PCRs would have been nice especially in the culture negative group. This might lend more credence to their theory that culture negative patients had less *c. difficile* burden.

With regards to their description of the two groups: It doesn't make sense that the patients did not differ in terms of severity of *c. difficile* or symptoms, though it is possible the groups were too small to show a statistically significant difference. It also doesn't make sense that they did not differ with regards to mortality, length of stay, or *C. difficile* recurrence. They really only differed with regards to how quickly they recovered from having diarrhea ie "response to therapy" as defined by authors, a definition that is subjective at best given the many reasons patients may have diarrhea in the hospital, beside just *C. difficile*.

I also found it fascinating that less than 50 percent of patients in each group received recommended therapy for *C. difficile*: (oral vancomycin or oral metronidazole, respectively), and that a large number of them received either probiotics or no therapy (30 percent in each group!). This is not representative, I would think of what happens in the rest of the world - and surely, this would impact the conclusion that culture negative group recovered from infection faster, because the treatments they received in almost half the cases were not standard of care, in either group. For the no therapy groups, the authors offer no explanation as to why that was - were these patients, then, not that symptomatic from diarrhea that they were not treated? Was their diarrhea not due to *C. difficile* in the first place? This then raises questions again about how they defined symptomatic *C. difficile* infection in the first place.

Statistical analysis was adequate and proper tests were employed for the data analysis. They did a thorough review of clinical presentation of cases and a nice description of underlying co-morbidities and risk factors for severe disease.

Third gen cephalosporins are known risk factor for severe c. difficile disease - there are known references to this in the literature, many of them, but none that I could find where third gen cephalosporins actually inhibit C. difficile growth, which would suggest a protective effect outside of the single study the authors cite. In fact, the Stiefel study (reference 34) that they do cite to support their conclusion actually is an animal study demonstrating the protective effect of gut anaerobes against C. difficile colonization, and it also reached the conclusion that treatment with a cephalosporin led to overgrowth of C. difficile in harvested cecal contents one day after inoculation (page 11 of 28 in this study). The study actually reached the same conclusion as other studies, namely that cephalosporins increase risk for C. difficile. The authors tried to use it to claim the opposite. Thus I think a different explanation should be offered as to why this association may have been found, or was it by chance?

In conclusion the authors showed considerable effort in characterizing clinical and microbiologic characteristics of these two groups of C. difficile patients, but there are significant concerns regarding their methodology, definition of positive C. difficile, and not enough evidence to support the conclusion that lack of growth in culture correlated with a difference in outcome.

Particular points for improvement:

1. What they did with discordant C. difficile testing results as outlined above
2. Cycle thresholds for PCRs in each of the two groups
3. Explanation as to why a significant number of patients did not receive standard treatment for c. difficile in each group - did they have non c. difficile related diarrhea?
4. Alternate explanation as to why prior cephalosporin therapy predisposed to lack of growth in culture (and also, please include actual patient numbers in table 3, not just odds ratios) given their conclusions are not supported by literature cited here.
5. Why was there a difference in success in therapy but no difference in terms of other hospital outcomes or recurrence rates? This is not discussed either.

Staff Comments:

Preparing Revision Guidelines

For complete guidelines on revision requirements, please see the Instructions to Authors at [link to page]. **Submissions of a paper that does not conform to Microbiology Spectrum guidelines**

will delay acceptance of your manuscript.

Please return the manuscript within 60 days; if you cannot complete the modification within this time period, please contact me. If you do not wish to modify the manuscript and prefer to submit it to another journal, please notify me of your decision immediately so that the manuscript may be formally withdrawn from consideration by Microbiology Spectrum.

If you would like to submit an image for consideration as the Featured Image for an issue, please contact Spectrum staff.

Spectrum 00799-21 Review.

I read with interest the manuscript submitted by Lee et al. The paper fits within the scope of microbiology spectrum and is of sound technical quality and of use to the community. However, there are a few concerns which I present below.

Major comments:

The inclusion criteria remain unclear. Where patients with positive growth on CCFA only included if they were toxin positive by PCR? Or did they also have to be GDH / toxin EIA positive? Unclear how the final group (n=158) was selected based on the laboratory work up. I would recommend authors to clarify this in the methods as this central to transparency and reproducibility.

Authors should include more detailed models about MV model build such as how variables were selected for inclusion (looks like by univariate p value cut offs?) where any other variables of clinical significance included? How was the model then assessed (one common method is by Hosmer–Lemeshow test, a statistical test for goodness of fit for logistic regression models)

Line 165: Were patients who received no active treatment excluded? Would include actual numbers with the percentages that are presented in the results (eg _% (_/_)). Would recommend restricting this comparison to patients who received active *C. difficile* treatment as recommended by IDSA guidelines (metronidazole and vancomycin). Similar to the subsequent MV model analysis.

The writing style makes the manuscript hard to comprehend and follow at time. This is most pronounced in the discussion section. Would recommend the authors review discussion section for clarity. It is customary to have the main findings of the paper summarized and put into context in the opening paragraph of the discussion. Would suggest the authors do this to increase readability of the discussion.

Minor comments

In figure one the framing of the exclusion criteria is unclear. Specifically, the wording “rely on

Methods: Was stool culture done as routine clinical work? Or was this done specifically as part of this study? Would recommend the authors clarify.

In the abstract and throughout the manuscript would consider rewording the phrase “no yielding “.

Line 88: Would recommend rewording this statement. It is unclear to me what the authors mean by “vast correlation between the positive yielding of *C. difficile* from

perianal samples and stool sample with the cycle thresholds(Ct) to PCR positivity". In the referenced study there was a weak correlation between peri-anal swab CT and PCR positivity and stool CT values and PCR positivity.

Line 207: Agree with the authors that the toxin EIA has unsatisfactory sensitivity as a stand-alone test for *C. difficile* diagnosis. However, with the advent and widespread use of NAAT toxins, I'm not sure there exist a role for routine *C. difficile* culture in the clinical settings and would recommend the authors reconsider this statement. Especially because the authors do not present any data in their own study to support this.

One limitation in the study is the omission of other antibiotic classes, including clindamycin which has a strong association with *C. difficile* occurrence and tetracyclines (eg: doxycycline or tigecycline) which may have protective function against *C. difficile* occurrence.

Given the link between toxin level and outcomes, could the findings here be driven by toxin level vs. mere *c. diff* burden as measured by ability to culture. This is not explored in the study although brought up in the discussion, and should likely be included as a limitation or expanded on in the discussion

Dear the editor of *Microbiology Spectrum*:

Enclosed, please find our revised manuscript entitled " **Clinical significance of the growth of toxigenic *Clostridioides difficile* in stool cultures during the era of non-culture methods for CDI diagnoses**". We have considered carefully for all concerns that were raised by the reviewers in the revised manuscript. The change was shown with "**red words**". We hope that our changes can clarify the points raised.

Best Regards,

Yuan-Pin Hung, Department of Internal Medicine, Tainan Hospital, Ministry of Health and Welfare, Tainan, Taiwan (yuebin16@yahoo.com.tw)

Wen-Chien Ko, Department of Internal Medicine, College of Medicine, National Cheng Kung University Hospital, Tainan, Taiwan (winston3415@gmail.com)

Reviewers' comments:

Reviewer comments:

Reviewer #1 (Comments for the Author):

I read with interest the manuscript submitted by Lee et al. The paper fits within the scope of microbiology spectrum and is of sound technical quality and of use to the community. However, there are a few concerns which I present below.

Major comments:

The inclusion criteria remain unclear. Were patients with positive growth on CCFA only included if they were toxin positive by PCR? Or did they also have to be GDH / toxin EIA positive? Unclear how the final group (n=158) was selected based on the laboratory work up. I would recommend authors to clarify this in the methods as this central to transparency and reproducibility.

Reply:

- 1. The PCR mentioned in this article was home-made multiplex PCR for detecting *tcdB* among *C. difficile* strains isolated, but not commercial PCR for detecting *tcdB* in fecal samples. (line 110)**
- 2. Since there were no commercial PCR for detecting *tcdB* in fecal samples available in our hospital, among patients with GDH positive but toxin test negative, subsequent diagnosis of CDI were confirmed by toxigenic *C. difficile* isolated, but they would not be included in this study. (line 102-104)**
- 3. Basically it is a retrospective chart review study, we reclarify the inclusion and exclusion criteria in our study as "Adults with CDI diagnosed by clinician based on positive for *C. difficile* toxin (by enzyme immunoassay or toxigenic stains isolated from culture) in fecal samples among patients with diarrhea between January 2013 and April 2020 were included. Patients without *C. difficile* GDH and toxin data, or no stool culture data were excluded. Among patients with GDH positive but toxin test negative, subsequent diagnosis of CDI were confirmed by toxigenic *C. difficile* isolated, but they would not be included in this study." (line 97-104).**
- 4. The procedure of patients selected in this study was clarified as "Among the retrospective chart review, 252 patients were diagnosed as CDI by clinician. Of the 252 patients with CDI, 43 were excluded, including 17 diagnosed reply on toxigenic *C. difficile* isolated without toxin test from fecal sample, 16 diagnosed rely on both**

positive GDH and toxin test from fecal sample without stool culture data; and 10 were GDH positive but toxin test negative, subsequent confirmed by toxigenic *C. difficile* isolated (Fig 1).” (line 143-147).

Authors should include more detailed models about MV model build such as how variables were selected for inclusion (looks like by univariate p value cut offs?) where any other variables of clinical significance included? How was the model then assessed (one common method is by Hosmer-Lemeshow test, a statistical test for goodness of fit for logistic regression models)

Reply

- 1. Parameters with *P* value less than 0.1 were selected for multivariate analysis. The multivariate analysis was performed with Hosmer–Lemeshow test for goodness of fit for logistic regression models. (line 138-140)**
- 2. Parameters, including prior ceftazidime or ceftriaxone therapy, having hypertension and chronic obstructive pulmonary disease with *P* value less than 0.1 were selected for multivariate analysis. Prior cefepime therapy, though with *P* value less than 0.1, might be confounding with another cephalosporin, prior ceftazidime or ceftriaxone therapy, so it was not selected for multivariate analysis. (line 170-174)**

Line 165: Were patients who received no active treatment excluded? Would include actual numbers with the percentages that are presented in the results (eg _% (_/_)). Would recommend restricting this comparison to patients who received active *C. difficile* treatment as recommended by IDSA guidelines (metronidazole and vancomycin). Similar to the subsequent MV model analysis.

Reply:

Of the 209 CDI patients included in this study, 114 (54.5 %) patients received active therapy, including metronidazole or vancomycin therapy according to the IDSA guidelines. (line 181-182). The therapeutic outcome of the 114 patients were re-calculated as following:

- (A) Among 114 patients with CDI receiving metronidazole or vancomycin therapy, patients with success in therapy comparing those without success, had higher rate of no *C. difficile* growth in stool cultures (30.5 vs. 3.1 %, *P*=0.001), higher proportions of comorbidities of hypertension (74.4 vs. 50.0 %, *P*=0.02), and older age (78.3±11.9 vs. 72.6±12.5 %, *P*=0.03), and less male gender (46.3 vs. 62.5 %, *P*=0.15) (Table 5). (line 182-186)**
- (B) In the multivariable analysis for the predictor of successful therapy among 114 patients receiving metronidazole or vancomycin therapy, successful therapy was**

positively associated with those without growth of *C. difficile* in stool (OR 12.70, CI 1.57-102.89, $P=0.02$) (Table 6). (line 187-189)

The writing style makes the manuscript hard to comprehend and follow at time. This is most pronounced in the discussion section. Would recommend the authors review discussion section for clarity. It is customary to have the main findings of the paper summarized and put into context in the opening paragraph of the discussion. Would suggest the authors do this to increase readability of the discussion.

Reply:

The opening paragraph of the discussion had been revised, including:

(A) The first paragraph of discussion was revised as “No growth of *C. difficile* from stool culture was associated with higher rates of successful CDI therapy in our study. The result might reasonably result from the lower *C. difficile* bacterial burden in stool.” (line 193-195)

(B) The second paragraph of discussion was revised as “Besides the advantage of predicting therapeutic outcomes of CDI noted in our study, stool culture, though time-consuming provide higher sensitivity compared with EIA in detecting the *C. difficile* toxins in fecal samples” (line 212-214)

Minor comments

In figure one the framing of the exclusion criteria is unclear. Specifically, the wording "rely on

Reply:

- 1. We re-clarify the inclusion and exclusion criteria of our study as “Adults with CDI diagnosed by clinician based on positive for *C. difficile* toxin (by enzyme immunoassay or toxigenic stains isolated from culture) in fecal samples among patients with diarrhea between January 2013 and April 2020 were included. Patients without *C. difficile* GDH and toxin data, or no stool culture data were excluded. Among patients with GDH positive but toxin test negative, subsequent diagnosis of CDI were confirmed by toxigenic *C. difficile* isolated, but they would not be included in this study.” (line 97-104)**
- 2. The descriptions in figure 1 was revised as “43 were excluded, including 17 diagnosed reply on toxigenic *C. difficile* isolated without toxin test from fecal sample, 16 diagnosed rely on both positive GDH and toxin test from fecal sample without stool culture data; and 10 were GDH positive but toxin test negative, subsequent confirmed by toxigenic *C. difficile* isolated.” (Figure 1)**

Methods: Was stool culture done as routine clinical work? Or was this done specifically as part of this study? Would recommend the authors clarify.

Reply:

- 1. It is a retrospective study. Fecal GDH/toxin test or stool culture were done based by clinician decision, not routine clinical work.(line 94-99)**
- 2. Therefore 16 patients had toxin test positive but no stool culture data. Though they were regarded as "having CDI", they would not be included for analysis. (line 145-146)**

In the abstract and throughout the manuscript would consider rewording the phrase "no yielding".

Reply:

We reworded the phrase "no yielding" as "no growth" in the abstract and throughout the manuscript

Line 88: Would recommend rewording this statement. It is unclear to me what the authors mean by "vast correlation between the positive yielding of C. difficile from perianal samples and stool sample with the cycle thresholds(Ct) to PCR positivity". In the referenced study there was a weak correlation between peri-anal swab CT and PCR positivity and stool CT values and PCR positivity.

Reply

The sentence was corrected as "there was a correlation....." (line 85-86)

Line 207: Agree with the authors that the toxin EIA has unsatisfactory sensitivity as a stand-alone test for C. difficile diagnosis. However, with the advent and widespread use of NAAT toxins, I'm not sure there exist a role for routine C. difficile culture in the clinical settings and would recommend the authors reconsider this statement. Especially because the authors do not present any data in their own study to support this.

Reply:

- 1. It was revised as "Besides the advantage of predicting therapeutic outcomes of CDI noted in our study, stool culture, though time- consuming provide higher sensitivity compared with EIA in detecting the C. difficile toxins in fecal samples" (line 212-214)**
- 2. Furthermore compared with real-time PCR in diagnosis CDI, culture-based method**

was found to detect an additional 9% of positive specimens, and most important result in 61% reduction in material costs (Shik Luk et al. Ref 33) (219-221)

One limitation in the study is the omission of other antibiotic classes, including clindamycin which has a strong association with *C. difficile* occurrence and tetracyclines (eg: doxycycline or tigecycline) which may have protective function against *C. difficile* occurrence.

Reply:

- 1. Clindamycin is not available in our hospital.**
- 2. Tigecycline or doxycycline therapy before CDI were recorded (line 123). There were no difference in age, gender, other underlying disease or prior antibiotic (including doxycycline and tigecycline) or medications exposure. (line 160-162)**

Given the link between toxin level and outcomes, could the findings here be driven by toxin level vs. mere *C. diff* burden as measured by ability to culture. This is not explored in the study although brought up in the discussion, and should likely be included as a limitation or expanded on in the discussion

Reply:

It was listed as limitation "Finally, ability to culture was suspected to be correlated with higher *C. difficile* burden in the study. Further study on the correlation between fecal *C. difficile* burden and fecal culture rate is needed to confirm our hypothesis. (line 243-246)

Reviewer #2 (Comments for the Author):

This is a well thought out study with an intriguing hypothesis that does indeed merit further study.

The authors' primary conclusion that the culture negative group responded more to therapy has some major flaws however.

The methodology description needs more detail, in order to lend credence to the conclusions which cannot be supported without additional information. The definition of positive *C. difficile* diagnoses is confusing. In the methods section, the authors outline that they used a multiplex PCR for the detection of toxin genes, but also a GDH and EIA assay for toxins. And yet this is not clarified in the flow chart in Figure 1 (what groups exactly did they include in their definition for toxin positive? What did they do with the discordants - PCR negative vs GDH/EIA positive or PCR positive vs GDH/EIA negative?). Correlation with cycle thresholds of

toxin PCRs would have been nice especially in the culture negative group. This might lend more credence to their theory that culture negative patients had less *C. difficile* burden.

Reply:

- 1. The PCR mentioned in this article was home-made multiplex PCR for detecting *tcdB* among *C. difficile* strains isolated, but not commercial PCR for detecting *tcdB* in fecal samples. (line 110)**
- 2. Since there were no commercial PCR for detecting *tcdB* in fecal samples available in our hospital, among patients with GDH positive but toxin test negative, subsequent diagnosis of CDI were confirmed by toxigenic *C. difficile* isolated, but they would not be included in this study. (line 102-104)**
- 3. Basically it is a retrospective chart review study, we reclarify the inclusion and exclusion criteria in our study as "Adults with CDI diagnosed by clinician based on positive for *C. difficile* toxin (by enzyme immunoassay or toxigenic stains isolated from culture) in fecal samples among patients with diarrhea between January 2013 and April 2020 were included. Patients without *C. difficile* GDH and toxin data, or no stool culture data were excluded. Among patients with GDH positive but toxin test negative, subsequent diagnosis of CDI were confirmed by toxigenic *C. difficile* isolated, but they would not be included in this study." (line 97-104).**
- 4. The procedure of patients selected in this study was clarified as "Among the retrospective chart review, 252 patients were diagnosed as CDI by clinician. Of the 252 patients with CDI, 43 were excluded, including 17 diagnosed reply on toxigenic *C. difficile* isolated without toxin test from fecal sample, 16 diagnosed rely on both positive GDH and toxin test from fecal sample without stool culture data; and 10 were GDH positive but toxin test negative, subsequent confirmed by toxigenic *C. difficile* isolated (Fig 1)." (line 143-147).**

With regards to their description of the two groups: It doesn't make sense that the patients did not differ in terms of severity of *C. difficile* or symptoms, though it is possible the groups were too small to show a statistically significant difference. It also doesn't make sense that they did not differ with regards to mortality, length of stay, or *C. difficile* recurrence. They really only differed with regards to how quickly they recovered from having diarrhea ie "response to therapy" as defined by authors, a definition that is subjective at best given the many reasons patients may have diarrhea in the hospital, beside just *C. difficile*.

Reply: we agree that "the CDI patients with or without stool culture positive for *C. difficile* did not differ in terms of severity of *C. difficile* or symptoms, mortality, length

of stay, or *C. difficile* recurrence. The result might be due to too small sample size to show a statistically significant difference.” Which was listed as limitations (line 246-249)

I also found it fascinating that less than 50 percent of patients in each group received recommended therapy for *C. difficile*: (oral vancomycin or oral metronidazole, respectively), and that a large number of them received either probiotics or no therapy (30 percent in each group!). This is not representative, I would think of what happens in the rest of the world - and surely, this would impact the conclusion that culture negative group recovered from infection faster, because the treatments they received in almost half the cases were not standard of care, in either group.

Reply:

Of the 209 CDI patients included in this study, 114 (54.5 %) patients received recommended therapy, including metronidazole or vancomycin therapy according to the IDSA guidelines. (line 181-182). The therapeutic outcome of the 114 patients were re-calculated as following:

(A) Among 114 patients with CDI receiving metronidazole or vancomycin therapy, patients with success in therapy comparing those without success, had higher rate of no *C. difficile* growth in stool cultures (30.5 vs. 3.1 %, $P=0.001$), higher proportions of comorbidities of hypertension (74.4 vs. 50.0 %, $P=0.02$), and older age (78.3 ± 11.9 vs. 72.6 ± 12.5 %, $P=0.03$), and less male gender (46.3 vs. 62.5 %, $P=0.15$) (Table 5). (line 182-186)

(B) In the multivariable analysis for the predictor of successful therapy among 114 patients receiving metronidazole or vancomycin therapy, successful therapy was positively associated with those without growth of *C. difficile* in stool (OR 12.70, CI 1.57-102.89, $P=0.02$) (Table 6). (line 187-189)

For the no therapy groups, the authors offer no explanation as to why that was - were these patients, then, not that symptomatic from diarrhea that they were not treated? Was their diarrhea not due to *C. difficile* in the first place? This then raises questions again about how they defined symptomatic *C. difficile* infection in the first place.

Reply:

About the 69 patients receiving no specific therapy, all had diarrhea cessation after stopping the offending antibiotics so clinician decided no specific therapy needed for CDI. The decision was compatible with the suggestions from IDSA/SHEA guidelines. (154-156).

Statistical analysis was adequate and proper tests were employed for the data analysis. They did a thorough review of clinical presentation of cases and a nice description of underlying co-morbidities and risk factors for severe disease.

Third gen cephalosporins are known risk factor for severe *C. difficile* disease - there are known references to this in the literature, many of them, but none that I could find where third gen cephalosporins actually inhibit *C. difficile* growth, which would suggest a protective effect outside of the single study the authors cite. In fact, the Stiefel study (reference 34) that they do cite to support their conclusion actually is an animal study demonstrating the protective effect of gut anaerobes against *C. difficile* colonization, and it also reached the conclusion that treatment with a cephalosporin led to overgrowth of *C. difficile* in harvested cecal contents one day after inoculation (page 11 of 28 in this study). The study actually reached the same conclusion as other studies, namely that cephalosporins increase risk for *C. difficile*. The authors tried to use it to claim the opposite. Thus I think a different explanation should be offered as to why this association may have been found, or was it by chance?

Reply:

We agreed with the uncertainty in this issue as described by the reviewer; so the sentence was revised as “The effect of disturbed microbiota in stool after third-generation cephalosporins exposure on decreasing the growth rate of *C. difficile* from stool cultures need further evaluation.” (line 230-232)

In conclusion the authors showed considerable effort in characterizing clinical and microbiologic characteristics of these two groups of *C. difficile* patients, but there are significant concerns regarding their methodology, definition of positive *C. difficile*, and not enough evidence to support the conclusion that lack of growth in culture correlated with a difference in outcome.

Particular points for improvement:

1. What they did with discordant *C. difficile* testing results as outlined above

Reply:

- 1. The PCR mentioned in this article was home-made multiplex PCR for detecting *tcdB* among *C. difficile* strains isolated, but not commercial PCR for detecting *tcdB* in fecal samples. (line 110-111)**
- 2. Since there were no commercial PCR for detecting *tcdB* in fecal samples available in our hospital, among patients with GDH positive but toxin test negative, subsequent diagnosis of CDI were confirmed by toxigenic *C. difficile* isolated, but they would not be included in this study. (line 102-104)**
- 3. Basically it is a retrospective chart review study, we reclarify the inclusion and**

exclusion criteria in our study as “Adults with CDI diagnosed by clinician based on positive for *C. difficile* toxin (by enzyme immunoassay or toxigenic stains isolated from culture) in fecal samples among patients with diarrhea between January 2013 and April 2020 were included. Patients without *C. difficile* GDH and toxin data, or no stool culture data were excluded. Among patients with GDH positive but toxin test negative, subsequent diagnosis of CDI were confirmed by toxigenic *C. difficile* isolated, but they would not be included in this study.” (line 97-104).

4. The procedure of patients selected in this study was clarified as “Among the retrospective chart review, 252 patients were diagnosed as CDI by clinician. Of the 252 patients with CDI, 43 were excluded, including 17 diagnosed reply on toxigenic *C. difficile* isolated without toxin test from fecal sample, 16 diagnosed rely on both positive GDH and toxin test from fecal sample without stool culture data; and 10 were GDH positive but toxin test negative, subsequent confirmed by toxigenic *C. difficile* isolated (Fig 1).” (line 143-147)

2. Cycle thresholds for PCRs in each of the two groups

Reply:

The PCR mentioned in this article was home-made multiplex PCR for detecting *tcdB* among *C. difficile* strains isolated, but not commercial real-time PCR for detecting *tcdB* in fecal samples. So there were no cycle thresholds for PCRs. (line 110-111)

3. Explanation as to why a significant number of patients did not receive standard treatment for *c. difficile* in each group - did they have non *c. difficile* related diarrhea?

Reply:

About the 69 patients receiving no specific therapy, all had diarrhea cessation after stopping the offending antibiotics so clinician decided no specific therapy needed for CDI. The decision was compatible with the suggestions from IDSA/SHEA guidelines. (154-156).

4. Alternate explanation as to why prior cephalosporin therapy predisposed to lack of growth in culture (and also, please include actual patient numbers in table 3, not just odds ratios) given their conclusions are not supported by literature cited here.

Reply:

1. We agreed with the uncertainty in this issue as described by the reviewer; so the

sentence was revised as “The effect of disturbed microbiota in stool after third-generation cephalosporins exposure on decreasing the growth rate of *C. difficile* from stool cultures need further evaluation.” (line 230-232)

2. The multivariate analysis in table 3 was for clinical predictors of 209 patients with CDIs mentioned in table 1 and table 2. To avoid the misleading, we clarify the sentence as “Parameters, including prior ceftazidime or ceftriaxone therapy, having hypertension and chronic obstructive pulmonary disease with *P* value less than 0.15 were selected for multivariate analysis. Prior cefepime therapy, though with *P* value less than 0.15, might be confounding with another cephalosporin, prior ceftazidime or ceftriaxone therapy, so it was not selected for multivariate analysis. Multivariate analysis for clinical predictors of 209 patients with CDIs,.....(Table 3)” (line 170-176)

5. Why was there a difference in success in therapy but no difference in terms of other hospital outcomes or recurrence rates? This is not discussed either.

Reply: we agree that “the CDI patients with or without stool culture positive for *C. difficile* did not differ in terms of severity of *C. difficile* or symptoms, mortality, length of stay, or *C. difficile* recurrence. The result might be due to too small sample size to show a statistically significant difference.” Which was listed as limitations (line 246-249)

August 7, 2021

Dr. Yuan Pin Hung
Tainan Hospital, Department of Health, Executive Yuan
Internal medicine
Tainan
Taiwan

Re: Spectrum00799-21R1 (Clinical significance of the growth of toxigenic *Clostridioides difficile* in stool cultures during the era of non-culture methods for CDI diagnoses)

Dear Dr. Yuan Pin Hung:

Thank you for submitting your manuscript to Microbiology Spectrum. As you can see, the reviewers have additional comments that should be addressed prior to acceptance, particularly clarifying the inclusion criteria and addressing effort and cost associated with performing *C. difficile* culture. In addition, we would recommend consulting a language editing service to assist with the concerns about grammar. A suggested list is available on the spectrum website at <https://journals.asm.org/language-editing-services>

When submitting the revised version of your paper, please provide (1) point-by-point responses to the issues raised by the reviewers as file type "Response to Reviewers," not in your cover letter, and (2) a PDF file that indicates the changes from the original submission (by highlighting or underlining the changes) as file type "Marked Up Manuscript - For Review Only". Please use this link to submit your revised manuscript - we strongly recommend that you submit your paper within the next 60 days or reach out to me. Detailed information on submitting your revised paper are below.

Link Not Available

Sincerely,

N. Esther Babady

Journals Department
Reviewer comments:

Reviewer #1 (Comments for the Author):

The reviewers have attempted to clarify and respond to reviewer comments. However, one major comment still remains.

Major comments:

The workflow for inclusion remains unclear. As it currently reads, it seems that inclusion criteria is based on GDH/toxin +ve EIA and presence of stool culture results only. However, in the methods the PCR is still mentioned. It is still unclear how the tcdB PCR factors in. The authors mention that if "Among patients with GDH positive but toxin test negative, subsequent diagnosis of CDI were confirmed by toxigenic *C. difficile* isolated, but they would not be included in this study." So seems like the PCR has no role in determining inclusion or exclusion in the cohort? This seems to be reflected in figure 1. If this is the case the PCR should not be mentioned in the methods at all as it is causing significant confusion.

Minor comments:

Line 144: "including 17 diagnosed reply on toxigenic *C.*" Unclear what reply here is intended to mean, the authors should double check this.

One limitation that should be addressed is lack of toxin quantification as the authors have outlined in their discussion how this could be a factor in *C. diff* outcomes

Reviewer #2 (Comments for the Author):

Thank you for addressing previously cited concerns. This version of the paper substantially explains the prior concerns regarding inclusion/exclusion criteria (Fig 1) and lack of standard treatments applied to roughly 50 percent of the patients and effect on their outcomes, while also better explaining limitations. I do think they can focus more on the cost-effectiveness of stool culture especially as this test applies in resource-limited settings such as their own, this will strengthen their argument in favor of stool cultures where PCRs are not as readily available. Grammar is still an issue - please review carefully as the grammar can make parts of the manuscript hard to understand; I understand English is not the author's first language though!

Staff Comments:

Preparing Revision Guidelines

For complete guidelines on revision requirements, please see the Instructions to Authors at [link to page]. **Submissions of a paper that does not conform to Microbiology Spectrum guidelines will delay acceptance of your manuscript.**

Please return the manuscript within 60 days; if you cannot complete the modification within this time period, please contact me. If you do not wish to modify the manuscript and prefer to submit it to another journal, please notify me of your decision immediately so that the manuscript may be formally withdrawn from consideration by Microbiology Spectrum.

If you would like to submit an image for consideration as the Featured Image for an issue, please contact Spectrum staff.

Dear the editor of *Microbiology Spectrum*:

Enclosed, please find our revised manuscript entitled "**Clinical significance of toxigenic *Clostridioides difficile* growth in stool cultures during the era of nonculture methods for the diagnosis of *C. difficile* infection**". We have considered carefully for all concerns that were raised by the reviewers. The manuscript was revised to a major extent in **the Abstract, Materials and Methods, and Results**, to improve the scientific logic and fluency of the whole manuscript. English editing has been done by an agent, American Journal Experts. The major changes were shown with "**red words**". We hope that our changes can clarify the points raised.

Best Regards,

Yuan-Pin Hung, Department of Internal Medicine, Tainan Hospital, Ministry of Health and Welfare, Tainan, Taiwan (yuebin16@yahoo.com.tw)

Wen-Chien Ko, Department of Internal Medicine, College of Medicine, National Cheng Kung University Hospital, Tainan, Taiwan (winston3415@gmail.com)

Reviewers' comments:

Reviewer #1 (Comments for the Author):

The reviewers have attempted to clarify and respond to reviewer comments. However, one major comment still remains.

Major comments:

The workflow for inclusion remains unclear. As it currently reads, it seems that inclusion criteria is based on GDH/toxin +ve EIA and presence of stool culture results only. However, in the methods the PCR is still mentioned. It is still unclear how the *tcdB* PCR factors in. The authors mention that if "Among patients with GDH positive but toxin test negative, subsequent diagnosis of CDI were confirmed by toxigenic *C. difficile* isolated, but they would not be included in this study." So seems like the PCR has no role in determining inclusion or exclusion in the cohort? This seems to be reflected in figure 1. If this is the case the PCR should not be mentioned in the methods at all as it is causing significant confusion.

Reply:

Materials and Methods - The first paragraph

- 1. To avoid misleading, we clarify the inclusion and exclusion criteria as “*Adult patients with unexplained diarrhea and positive EIA results of GDH and toxin A/B in fecal samples and available stool culture and multiplex PCR results were included in the present study. Patients without GDH/toxin or C. difficile stool culture data were excluded.*” in line 105-108. In another word, PCR was used to confirm the carriage of *tcdB* gene in the stool isolates, and had a limited role to include or exclude the study patients, since all the *C. difficile* isolates obtained from the stool samples with positive results of GDH and toxin A/B EIA toxigenic.**
- 2. The definition of CDI, the laboratory testing for GDH and toxin A/B in stool samples, and the brief description of multiplex PCR were re-organized in line 95-105.**

Figure 1 – It was modified to match the above revisions, esp. for the reasons of the exclusion of 43 patients.

Results – The inclusion of 209 patients who met the inclusion criterion of the presence of both GDH and *C. difficile* toxin A/B detected by EIA in stool explained the study population (lines 140-145).

Minor comments:

Line 144: "including 17 diagnosed reply on toxigenic *C.*" Unclear what reply here is intended to

mean, the authors should double check this.

Reply: As the above reply, the definition of inclusion criteria was revised to make the study population clarified. In figure 1, those 17 patients with toxigenic *C. difficile* isolated detected in stool culture, but without EIA toxin data, were excluded in the study.

One limitation that should be addressed is lack of toxin quantification as the authors have outlined in their discussion how this could be a factor in *C. diff* outcomes

Reply: As the reviewer's suggestion, no quantification of fecal toxin level was added as one of the limitation in line 242-245: "*Finally, fecal C. difficile toxin was qualitatively detected using a commercial EIA kit. The quantified measurement of C. difficile toxin was not available in our clinical laboratory, although fecal C. difficile toxin levels are often referred to as one of the prognostic factors of CDI (26-29).*"

Reviewer #2 (Comments for the Author):

Thank you for addressing previously cited concerns. This version of the paper substantially explains the prior concerns regarding inclusion/exclusion criteria (Fig 1) and lack of standard treatments applied to roughly 50 percent of the patients and effect on their outcomes, while also better explaining limitations. I do think they can focus more on the cost-effectiveness of stool culture especially as this test applies in resource-limited settings such as their own, this will strengthen their argument in favor of stool cultures where PCRs are not as readily available.

Reply: The sentences to highlight the cost-effectiveness of stool cultures, if PCR tests are not available, were added in line 215-223: "*Although real-time PCR has been suggested as the confirmatory tool when the results of GDH and toxin EIA tests are inconsistent, according to the IDSA/SHEA guidelines, the higher cost of commercial real-time PCR tests means that the tests are not often available in resource-limited settings. Stool cultures are more cost-effective than commercial real-time PCR tests and could be applied as a complementary test in the presence of discordant results of GDH and toxin EIA testing. Accordingly, we believe that in the modern era of nonculture methods for the clinical diagnosis of CDI, the yields of toxigenic stool cultures are shown to have prognostic significance in the setting of conventional antimicrobial therapy for CDI.*"

Grammar is still an issue - please review carefully as the grammar can make parts of the manuscript hard to understand; I understand English is not the author's first language though!

Reply: The manuscript has been edited for English grammar by the American

Journal Experts with the verification code 8681-5097-5B44-7655-2630, as shown as follows.

Editing Certificate

This document certifies that the manuscript

Clinical significance of toxigenic Clostridioides difficile growth in stool cultures during the era of non-culture methods for the diagnoses of C. difficile infection

prepared by the authors

Ching-Chi Lee, Jen-Chieh Lee, Chun-Wei Chiu, Pei-Jane Tsai, Wen-Chien Ko and Yuan-Pin Hung

was edited for proper English language, grammar, punctuation, spelling, and overall style by one or more of the highly qualified native English speaking editors at AJE.

This certificate was issued on **August 15, 2021** and may be verified on the [AJE website](https://aje.com) using the verification code **8681-5097-5B44-7655-2630**.

Neither the research content nor the authors' intentions were altered in any way during the editing process. Documents receiving this certification should be English-ready for publication; however, the author has the ability to accept or reject our suggestions and changes. To verify the final AJE edited version, please visit our verification page at aje.com/certificate. If you have any questions or concerns about this edited document, please contact AJE at support@aje.com.

AJE provides a range of editing, translation, and manuscript services for researchers and publishers around the world. For more information about our company, services, and partner discounts, please visit aje.com.

September 7, 2021

Dr. Yuan Pin Hung
Tainan Hospital, Department of Health, Executive Yuan
Internal medicine
Tainan
Taiwan

Re: Spectrum00799-21R2 (Clinical significance of toxigenic *Clostridioides difficile* growth in stool cultures during the era of nonculture methods for the diagnosis of *C. difficile* infection)

Dear Dr. Yuan Pin Hung:

Your manuscript has been accepted, and I am forwarding it to the ASM Journals Department for publication. You will be notified when your proofs are ready to be viewed.

Sincerely,

N. Esther Babady
Editor, Microbiology Spectrum
